# Current Status of Cardiac Regenerative Therapy Using Induced Pluripotent Stem Cells

**DOI:** 10.3390/ijms25115772

**Published:** 2024-05-26

**Authors:** Tadahisa Sugiura, Dhienda C. Shahannaz, Brandon E. Ferrell

**Affiliations:** Department of Cardiothoracic and Vascular Surgery, Montefiore Medical Center/Albert Einstein College of Medicine, New York, NY 10467, USA; dhiendaladdynasrul@gmail.com (D.C.S.); bferrell@montefiore.org (B.E.F.)

**Keywords:** cardiac regenerative therapy, induced pluripotent stem cell, stem cell therapy, heart failure

## Abstract

Heart failure (HF) is a life-threatening disorder and is treated by drug therapies and surgical interventions such as heart transplantation and left ventricular assist device (LVAD). However, these treatments can lack effectiveness in the long term and are associated with issues such as donor shortage in heart transplantation, and infection, stroke, or gastrointestinal bleeding in LVADs. Therefore, alternative therapeutic strategies are still needed. In this respect, stem cell therapy has been introduced for the treatment of HF and numerous preclinical and clinical studies are employing a range of stem cell varieties. These stem cells, such as embryonic stem cells (ESCs) and induced pluripotent stem cells (iPSCs), have been shown to improve cardiac function and attenuate left ventricular remodeling. IPSCs, which have a capacity for unlimited proliferation and differentiation into cardiomyocytes, are a promising cell source for myocardial regeneration therapy. In this review, we discuss the following topics: (1) what are iPSCs; (2) the limitations and solutions for the translation of iPSC-CMs practically; and (3) the current therapeutic clinical trials.

## 1. Introduction

HF following myocardial infarction—ischemic cardiomyopathy—is a major cause of death and disability worldwide [1]. It is responsible for 1–2% of the total healthcare expenditure in industrialized countries, and has estimated annual direct costs reaching $70 billion in American society, impacting 6 million Americans. Globally, there are 65 million people who are affected by HF [2,3]. HF events have increased in the past 30 years [4] and are expected to increase at least until 2050, with the global expectation of a “heart failure pandemic”, as it is the second most common inpatient diagnosis billed to Medicare [5,6]. Myocardial infarction is a condition of acute injury to myocardium which kills as many as 25% of cardiomyocytes (CMs) in the ventricles [7]. The structural alteration will lead to more intensified mechanical stress of the ventricular wall to compensate for where contractile dysfunction has worsened, leading to HF. The extent of this heart failure condition is directly correlated to the amount of cardiomyocyte loss [8].

Ischemic cardiomyopathy accounts for ~50% of heart failure with reduced ejection fraction (HFrEF) and ~25% of all HF [9]. While existing pharmacological and device therapies improve functional status and reduce mortality in this syndrome, many patients progress to advanced HF and death [10]. Cardiac transplantation can be an effective treatment, but it is limited by organ donor availability [11]. To fill this therapeutic gap, stem cell therapy has recently emerged in the treatment of HF, and numerous preclinical and clinical studies using various types of stem cells have been shown to improve cardiac function and attenuate LV remodeling. Wherein ESCs are derived from preimplantation embryos, for which some ethical considerations of blastocyst destruction at the 6th–8th day of human embryo development arise [12], iPSCs are produced by the epigenetic reprogramming of somatic cells, such as skin fibroblasts or mononuclear blood cells. Moreover, iPSCs have been proven to possess a capacity to derive functional cardiomyocytes. Thus, the scalable production of iPSCs has become more favorable in the current cardiac regenerative medicine [13,14].

## 2. Induced Pluripotent Stem Cell (iPSC)

In 2006, Yamanaka succeeded in reprogramming fully matured somatic cells into pluripotent stem cells using four reprogramming factors, which is now famously known as “Yamanaka Factors”. These four factors are: Octamer-binding Transcription Factor 4 (OCT4), Sex-determining Region Y-box 2 (SOX2), Krüppel-like Factor 4 (KLF4), and c-Myc protein [15,16]. The resultant stem cells from this reprogramming are now called iPSCs, which are reprogrammed somatic cells with cell differentiation capacity to all the three embryonic germ layers [17]. The main idea behind the development of iPSCs is that overexpression of the genes which maintain pluripotency are sufficient to reprogram a somatic cell into an embryonic stem cell-like state [15]. These cells can then be differentiated into almost any cell of interest for use in regenerative therapies (Figure 1) [18]. The character properties of these cells are similar in that they offer long-term proliferation with the capacity to expand into iPSC-CMs. Then, these cells differ due to specific engineered differences formed using Clustered Regularly Interspaced Short Palindromic Repeats (CRISPR)/CRISPR associated protein 9 (Cas9) technology, which in the future can provide more mechanical profile insights into genetic disease. Moreover, the facts will help enable further development of precision medicine for drugs, biomarkers, and diagnostic purposes, with a patient-centered approach.

The advantages of an iPSC-based approach include an unlimited source of replacement cells and the avoidance of potential controversy concerning the use of fetal tissue [19]. Another advantage of undifferentiated iPSCs is that they can be cryo-banked in large numbers, which enables genetically identical seed stocks to be differentiated into CMs. This overcomes issues regarding genetic variation between different individuals and also makes it possible to create prescribed, large-scale banks of iPSCs with desired genotypes such as disease classes and genetic groups. Moreover, microarray research has demonstrated that hundreds of genes as well as DNA methylation patterns, are expressed differently between iPSCs and ESCs [20,21,22], showing the cruciality of cell derivation being used for the success of cell regeneration. Measurement of a range of properties of iPSCs and ESCs, such as gene expression, DNA methylation, microRNA expression, differentiation propensity, and complement activities which have been found in embryos suggest that the properties do vary [23]. Additionally, with different kinds of Cas9/CRISPR configurations, recent science makes it possible to create iPSC-derived CMs (iPSC-CMs) carrying targeted gene knockout or knock-ins and polymorphic substitutions, whilst the activation of certain genes, their repression of expression, and epigenetic regulation is also feasible [24]. This enables certain polymorphisms’ effects on cell function to be assessed in an otherwise constant genome (isogenic), which is crucial as the genetic background of phenotypes can exceed the mutations. Since its advent, the utilization of iPSC-CMs has revolutionized cardiology research, offering a versatile tool for investigating cardiac function and disease mechanisms at both in vitro and in vivo levels. In in vitro, iPSC-CMs enable the modeling of cardiac diseases, drug screening and toxicity testing, allowing for the identification of novel therapeutic targets and the development of personalized, precision medical approaches in regenerative medicine (Figure 1). Meanwhile, in vivo studies have shed new light for research advancements in the utilization of iPSCs in CM regeneration, which can potentially affect human lifespan by way of extension and productivity. Thus, iPSCs became a landmark development in regenerative medicine tested in vivo [25].

### 2.1. Disease Modeling

Within in vitro investigations, iPSC-CMs have pushed the modeling of cardiac disorders in many ways, such as by assessing the mitochondrial maturation process within hypertrophic cardiomyopathy [26], ion-mutation protein-derived arrhythmias in Long QT Syndromes [27], and other channelopathies found in dilated cardiomyopathy [28]. These can ultimately lead to genome-editing strategies becoming the major goal in solving cardiac diseases.

### 2.2. Drug Screening

For drug screening, iPSC-CMs help in preclinical process by confirming the drug effects on action potentials or ionic currents in iPSC-CMs [29,30]. The method has been used in the development process of the antiarrhythmic drugs, Dofetilide [31] and Mexiletine [32]. Another drug screening was done and proven in the Valtrate-induced myocardial strain [33].

### 2.3. Regenerative Therapies

iPSC-CMs have also emerged as a promising tool for regenerative therapies aimed at repairing or replacing damaged cardiac tissue by in vivo investigations. For instance, iPSC-CMs have been investigated for their potential to repair myocardial infarction, and studies have also been undertaken demonstrating their ability to differentiate between functional CMs and improve cardiac function in animal models. Additionally, iPSC-CMs have been explored for their capacity to generate bioengineered cardiac tissues, such as cardiac patches and bioartificial hearts, which can be used to replace damaged or diseased cardiac tissue. This will be further discussed in the following sections.

## 3. Limitations and Solutions for Translation of iPSC-CMs Practically

Therapy using iPSC-CMs has drawn global attention as a potential therapy for treating different kinds of heart disease. However, the idea of bringing the invention into mass production of iPSC-CMs remains challenging, thus limiting the potential in regenerative medicine. There are still some obstacles in the scalable production of iPSC-CMs. Based on the stage of process, these obstacles are: (1) Immaturity of iPSC-CMs, (2) Delivery routes, (3) Poor engraftment of implanted cells, (4) Immunogenicity and immune rejection, (5) Arrhythmogenicity, and (6) Possible tumorigenesis.

### 3.1. Overcoming iPSC-CMs Immaturity

iPSC-CMs still exhibit immature morphology and are similar to fetal CMs. The morphology is rounded, smaller than adult CMs, and mostly mononuclear. In contrast, adult CMs tend to be rod-shaped, anisotropic, and have a high aspect ratio, with 25–30% binucleated [34,35]. Whereas sarcomere is the essential contractile unit of CMs necessary for cardiac function, iPSC-CMs lack clear T-tubes and show disordered sarcomere stripes, which are shorter in length (1.6 µm), contain irregular myofibril, and most cells only have immature high-density Z-band and I-band [36].

Mature iPSC-CMs are characterized by mitochondrial maturation, increased oxidative capacity, and enhanced fatty acid use for energy production [26]. As the physiological hypertrophy caused by the organization of sarcomere structures contributes to the process of iPSC-CMs maturation, so is the development of T-tubules, which associates with the transition to mature CMs’ pattern of excitation-contraction coupling, synchronously increasing Ca^2+^ across cells. Thus, the iPSC-CM maturation process must involve more efficient calcium handling as well as improved electrophysiological properties causing higher force of contractility. This shows that CMs with properties that more closely resemble adult myocardium would reduce arrhythmia risk and have improved contractile properties when transplanted. Investigations into achieving such a goal (iPSC-CM maturation) has been done with several methods. Trials include long period culture (expected to last 80–120 days) [37], alteration to the culture’s substrate stiffness [38], electrical stimuli [39], and biochemical properties [40]. Mechanical loading has also been used to stimulate maturation [41]. Additionally, tissue engineering methods and electrical training of iPSC-CMs in a three-dimensional culture system has also contributed to the advancement of morphologically matured iPSCs [42]. Lastly, a 3D culture containing multiple cell types has also been investigated for the promotion of more developed iPSC-CM phenotypes [43].

### 3.2. Delivery Routes

Previous studies used isolated floating CMs for transplantation. The method involves trypsin, in which it helps with cell degradation, cell-adhesion molecule on cell-surface, and many growth factor (GF) receptors and ion channels. Moreover, there is a method found that also helps improve transplantation success rate, which is by freezing CMs. This method is found to lower cell engraftment rates by 5% after transplantation [44]. This low rate leads to inflammatory cell infiltration that works by removing necrotic cells that fail to conduct angiogenesis. As the transplanted tissue on the epicardial surface fails to connect to host CMs, its beating will be unsynchronized with that of host CMs. To solve this problem, treatment without trypsin was needed, one of them being CM Spheroids [44]. A CM spheroid is a 1000-CM aggregate with some advantages that include: the cells harvesting without trypsinization, the protection of cell-surface proteins from being damaged, retained extracellular matrix, and intact humoral factors (such as GF). Hence, CM spheroid increases cell-survival rate after transplantation remarkably.

Cardiomyocyte sheet, patch, or spheroid engraftment need a transplantation device that would create a hospitable environment for regenerating myocardium to sustain in a stable condition for a long period of time. If the implantation process does not achieve this, most of the cells are not retained in the myocardium. Furthermore, since the blade tip of a normal injection needle would cause bleeding that interferes with the process of transplantation, recent research has found that an injection needle with a cone-shaped blind end without a blade at the tip would reduce the bleeding interference that is otherwise caused by a conservative blade [45]. It has six holes on the side for injecting CM Spheroids into heart tissue. Engraftment would be easier to achieve success with this innovation. Still, there are rooms for improvements so the transplanted cells can enlarge physiologically and begin to resemble adult CMs by longitudinal elongation morphologically [46].

### 3.3. Prevention of Immunogenicity

The immunological barrier is one of the major hurdles in stem cell therapy, including iPSCs. The transplantation from allogeneic cells or tissues often elicits immune responses that eventually lead to graft rejection. In reality, immune rejection is a tandem reaction produced by immune cells attacking the graft through cytokine release, inflammation process, cytotoxic mechanism, and phagocytosis [47]. The process of immune rejection is divided in two stages: the sensitization and the effector stage. The first stage, sensitization of the lymphocyte of the recipient, identifies the antigen of the graft, then it activates and proliferates. In the effector stage, the graft is entailed to destruction. In the process, it might have harmful consequences for the transplant recipient. The rejection depends on how far organs are involved, which is initiated by T cells (that also plays the role of adaptor to secondary lymphocytes). Acute rejection starts within the first weeks after the transplant procedure, typically caused by syngeneic grafts, and usually happens within 10 days of the transplantation process, resulting in tissue destruction by macrophages and lymphocyte infiltration. In humans, T-cell mediated hypersensitivity and cytotoxicity play important roles in graft rejection. Solutions to the issue usually comprises major histocompatibility (MHC)-matching and the production of iPSC banks for options, autologous iPSCs that offer potential immune privilege, to the idea of a knockout to human leukocyte antigens (HLA) that may generate ready-to-use immunocompatible cells.

iPSCs carrying knockout mutations for two key components (β2 microglobulin and class II MHC class transactivator) of MHC I and II (i.e., HLA I/II knockout iPSCs) were generated using the CRISPR/Cas9 gene editing system and differentiated in CMs. The study is still in progress, but gives promise to its suitability for allogeneic transplantation inducing little to no activity in human immune cells [48].

In MHC matching, Shiba investigated a cynomolgus macaque with MI, in which allogeneic non-human primate iPSC-CMs were transplanted 14 days after injury [49]. Improved cardiac function, electrical coupling synergized with the host’s myocardium, and no evidence of immune rejection in the MHC-matched iPSC-CM trial group were found, with the only suggestion being that MHC-matched donor-derived iPSC-CMs transplantation would potentially promise safety if tested in humans. With autologous iPSCs, theoretically it should be able to avoid immune rejection, however, genetic and epigenetic alterations may also occur during reprogramming, so there is still a fair risk of immune rejection with autologous iPSCs transplantation [47]. Moreover, it takes several months to make autologous iPSCs as it requires the step of collecting the patient’s cells and generating CMs at a large scale. As it is highly costly and takes a long duration, it may not be feasible in current clinical practice.

### 3.4. Solutions for Arrhythmogenesis

Previous investigations of iPSC-CM transplantation showed that intramyocardial engraftment into non-human primates and porcine models was associated with the events of transient ventricular arrhythmia. Conductive scaffolds are then proposed to aid signal propagation [19]. Furthermore, engrafted iPSC-CMs tend to have immature phenotypes associated with spontaneous beatings, affecting cardiac electrical signaling. To reduce arrhythmia events, protocols on subtype-specific ventricular differentiation combined with iPSC-CMs maturation methods would be efficient to eliminate pacemaker-like activity elicited by engrafted cells [44].

### 3.5. Solutions to Poor Engraftment

The low rate of cell survival and retention after transplantation has been the central barrier in the advancement of effective iPSC-CM-based cardiac regenerative therapy. To improve the intramyocardially injected iPSC-CM survival rate, a pro-survival cocktail injection was developed to address graft deaths [50]. In epicardial patches, pre-vascularization strategies have been explored to promote anastomosis of the patches with the host vasculature and promote cell survival [51].

In intramyocardial injection of iPSC-CMs, fibrosis develops around the transplanted iPSC-CMs as a wound healing response from MI condition and the injection itself. This is the second problem, aside from the host’s immune response. It has been reported that less than 10% of injected cells successfully engraft and survive after its cardiac delivery [52]. Given how transplanted CMs die from ischemia in the first few days following the transplantation procedure, it is important to create a favorable environment by enhancing local vascularization to promote the cell’s survival rate [53]. If not, this poor engraftment causing ischemia of the cell affects the graft’s signal propagation and proper electromechanical integration leading to arrhythmia. Thus, the inventions to solve this issue include the use of overexpression of certain proteins, such as Cyclin D2 (which is involved in cell cycle regulation) [54], N-cadherin (a protein responsible for cell adhesion) [55], and co-transplanting readily made microvessels obtained from adipose tissue [19]. All result in increased graft perfusion, adhesion, and cell survival rate, leading to improved functional recovery following myocardial infarction [56].

### 3.6. Prevention of Tumorigenesis

Although iPSCs have a high growth rate and pluripotency suitable for regenerative medicine material, not all the cells cultured from iPSCs can differentiate into targeted cells. The remaining nontarget cells and the undifferentiated cells could form a tumor called a teratoma, which would impede clinical application. Previous studies have indicated that the iPSC tumorigenic potential is related to the C-Myc oncogene expression and the p53 genotype status of the cells [57,58]. As mentioned earlier, C-Myc is one of the Yamanaka Factors, the origin of somatic tissues from which pluripotent stem cells are induced. However, the thorough molecular mechanism behind iPSC tumor potentiality has to be further elucidated.

Some studies use purification stages to eliminate redundant non-CM cells as another method to prevent tumorigenesis. To solve this teratogenic tendency, studies have performed detailed analysis of energy metabolism-related pathways of CMs and iPSCs, which lands researchers on some solutions where lactate (which is the substance used for uptakes by CMs alongside glucose and glutamine) is added in the culture medium of iPSCs (Figure 2) [44]. Although iPSCs use glutamine and glucose as their main energy sources, CMs use lactate as an additional energy source. In a culture medium without glutamine and glucose, but containing lactic acid, only CMs can survive. This purification process by adding lactate is in recent findings of iPSCs exploration. This new method results in CMs remaining active with undifferentiated iPSCs and the non-CM count reduced to 0.001% in detection sensitivity; hence, teratoma formation did not occur.

## 4. Current Therapeutic Clinical Trials

Various clinical trials using iPSC-CMs are ongoing worldwide, but the majority are nontherapeutic studies, such as disease modeling, drug screening, or cell banks [59]. We introduce four therapeutic clinical trials of cardiac regenerative therapy using iPSC-CMs currently in progress (Table 1).

### 4.1. iPSC-Derived Cardiomyocyte Patch

Cuorips Inc. (Osaka, Japan) and a surgical group in Japan have explored a new strategy of myocardial regeneration therapy using the iPSC-CMs and cell-sheet technique. Cell-sheet technology involves utilizing poly (N-isopropylacrylamide) PNIPAAm (a thermo-responsive polymer) as a coat for culture dishes. This technique allows cells to be released from the dish and produces cell-sheets following temperature change and fabricates a scaffoldless cardiomyocyte patch from iPSCs [60]. Preclinical studies with swine myocardial infarction models have demonstrated their therapeutic potential [61,62,63]. The group has been conducting a clinical trial implanting an iPSC-CM patch in patients with end-stage ischemic cardiomyopathy (Trial ID: NCT04696328) and recently reported one-year outcomes [64]. The inclusion criteria were a left ventricular ejection fraction (LVEF) of 35% or less and New York Heart Association (NYHA) class III or higher heart failure symptoms. Three iPSC-CM patches, with 3.3 × 10^7^ cells per patch, about 3.5 cm in diameter, were transplanted on the LV anterior and lateral wall by left thoracotomy. In the first three cases of the trial, no transplanted cell-related adverse events were observed during the one-year observation period. Improvements in LV contractility by echocardiogram, electrocardiogram-gated cardiac CT, and an improvement in myocardial blood flow neovascularization by NH3-PET/CT were observed in two of the three patients. There was no evidence of tumor formation detected by whole-body FDG-PET scan or in surveillance blood testing of four biomarkers (alpha-fetoprotein, carbohydrate antigen 19-9, carcinoembryonic antigen, and human chorionic gonadotropin) after transplantation. In this clinical trial, no immunosuppressive agents were administered three months after the transplantation of the allogeneic iPSC-CM patches without matching HLA typing since the transplanted cells were expected to have been lost within the first three months. The paracrine effect is considered the primary mechanism of improvement in cardiac function, which leads to the improvement in myocardial blood flow by angiogenesis and the functional improvement of residual cardiomyocytes.

### 4.2. Cardiac Spheroid

Heartseed Inc. (Tokyo, Japan) has been conducting a phase I/II clinical study (LAPiS Study: jRCTa032200189) with a cardiac spheroid which is an aggregation of allogeneic iPSC-derived highly purified ventricular CMs used on patients with ischemic cardiomyopathy. Highly purified (claimed to be >99%) ventricular CMs decrease the presence of arrhythmia after transplantation of iPSC-CMs [65,66]. Fukuda, who leads the Heartseed, is stated to have developed methods to safely and efficiently generate iPSCs from peripheral blood T cells, then generate it into types of ventricle-specific CMs, distinguish the high quality iPSCs from its residual iPSCs using the “metabolic selection method”, and getting the iPSCs purified from its derivatives. The team also claimed to have developed a technology by which they can mass-produce the engraftment of CMs efficiently using cardiac spheroids. By forming micro-tissue-like spheroids, the retention rate and viability of transplanted cells are improved. One cardiac spheroid contains approximately 1000 CMs, and its diameter is about 200 µm. The LAPiS Study is a 52-week, phase I/II, dose-escalation clinical study in patients with ischemic cardiomyopathy which is being conducted in Japan. The primary endpoint of the study is safety at 26 weeks post-transplantation, and secondary efficacy endpoints include LVEF and myocardial wall motion. In this study, the cardiac spheroids are transplanted using a special administration needle into the LV myocardium in combination with coronary artery bypass grafting (CABG) surgery. The expected mechanism of action is that the transplanted CMs electrically couple with the patient’s myocardium to improve cardiac output by remuscularization and secretion of angiogenic factors (angiogenesis). They have recently reported on the 6-month outcomes of the first two patients in this study. They reported that the LVEF improved and there was a decreased left ventricular end-diastolic volume (LVEDV) by echocardiogram and cardiac MRI in both patients. There was also improvement in the NYHA functional class by one or two class reductions and a decrease in NT-proBNP in both patients. The study will enroll ten patients in two dose cohorts of 50 million and 150 million CMs.

### 4.3. Biological Ventricular Assist Tissue (BioVAT)

Repairon (Goettingen, Germany) has been conducting a Phase I/II trial (NCT04396899) in Germany evaluating the safety and efficacy of iPSC-derived Engineered Human Myocardium (EHM) as Biological Ventricular Assist Tissue (BioVAT) in end-stage heart failure [67]. The EHM were constructed from iPSC-CMs and stromal cells suspended in a bovine collagen type I hydrogel. EHM are produced from a comprehensively characterized iPS-cell line for off-the-shelf administration as allografts under concomitant immune suppression. EHM allografts can be produced on stock and delivered as off-the-shelf products for individual assembly to meet clinical dosing and size demands. EHM is administered by minimally invasive surgery onto the beating heart with an elective left-lateral mini thoracotomy. Zimmermann recently reported the latest outcomes of the BioVAT. The study demonstrated: (1) safe maximal dose of EHM grafts constructed from 800 million iPSC-CMs and stromal cells with 12 months follow-up, (2) proof of principle for vascularized remuscularization of the heart, (3) evidence for sustainable thickening of the target heart wall by EHM grafts, and (4) evidence for improved EF and symptoms in NYHA and the Kansas City Cardiomyopathy Questionnaire (KCCQ).

### 4.4. Epicardial Injection of iPSC-CMs

Wang et al. reported that two patients with ischemic cardiomyopathy underwent epicardial injection of allogeneic iPSC-CMs at the time of CABG in China under a clinical trial (NCT03763136) [68]. This study is a dose-escalation, placebo-controlled, single-center phase I/IIa clinical trial, in which the dose escalation is for three doses (1 × 10^8^, 2 × 10^8^, and 4 × 10^8^ cells) [69]. Patients with advanced heart failure are randomly allocated to receive epicardial injection of iPSC-CMs during CABG surgery or CABG surgery alone, followed by a 12-month follow-up investigation. The primary endpoint is to assess the safety of iPSC-CMs transplantation, including hemodynamic compromised ventricular arrhythmias and newly formed tumors during the initial six months postoperatively. The secondary endpoint is to evaluate the efficacy of the epicardial injection of iPSC-CMs and CABG surgery combination in comparison to isolated CABG surgery. The outcomes have not been reported yet.

## 5. Summary

The transplantation of iPSC-CMs represents an emerging therapeutic option for patients with end-stage HF and is keenly anticipated. Although clinical translation of iPSC-CMs therapy still has several limitations, future investigation of these barriers to translation should be conducted to push findings forward. Transplantation of iPSC-CMs is one of the most promising cardiac regenerative therapies and several clinical trials are currently in progress worldwide. It could potentially provide a treatment for end-stage HF by supplementing matured cardiomyocytes.

## Figures and Tables

**Figure 1 ijms-25-05772-f001:**
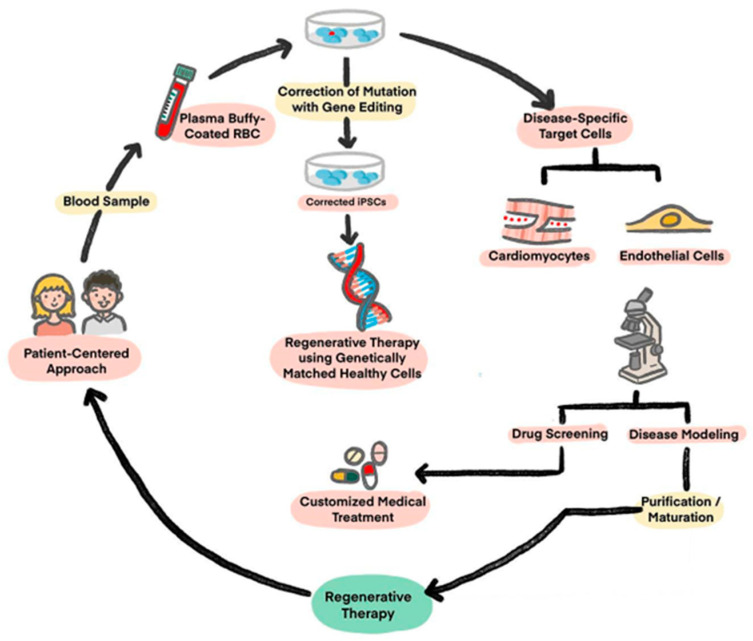
Potential application of iPSCs: Derivation, engineering, and utility algorithm of iPSCs. (Figure is adapted from Reference [18], Figure 1).

**Figure 2 ijms-25-05772-f002:**
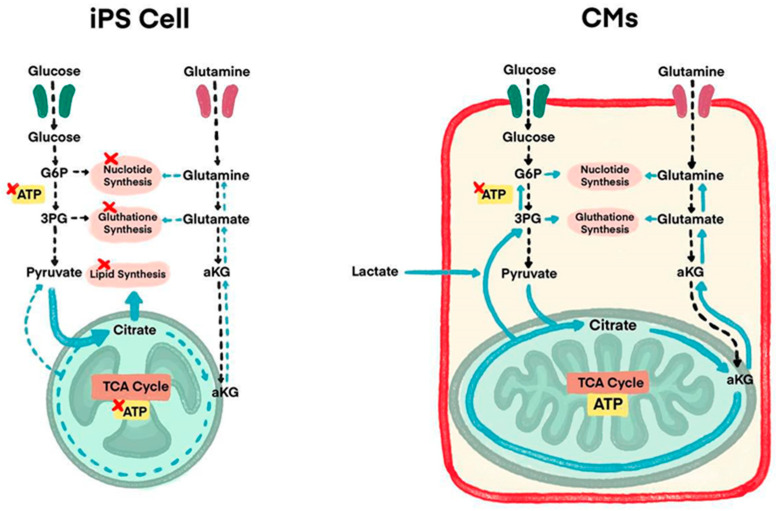
Understanding energy metabolism differences between iPSCs and CMs to overcome immaturity issues and tumorigenesis by metabolic maturation (Figure is adapted from Reference [44], Figure 1).

**Table 1 ijms-25-05772-t001:** List of therapeutic clinical trials using iPSC-CMs. RCT: randomized clinical trial, LV: left ventricle, LVEFL: left ventricular ejection fraction, iPSC-CM: induced pluripotent stem cell-derived cardiomyocyte, CABG: coronary artery bypass grafting, NYHA: New York Heart Association, HFrEF: heart failure with reduced ejection fraction.

Clinical Trial	Sponsor	Study Title	Inclusion	Intervention	Sample Size	Start Date	Country
NCT 03763136 Single-Blind RCT Phase I/IIa	Help Therapeutics, Nanjing University Medical School	Epicardial injection of allogeneic human pluripotent stem cell- derived cardiomyocytes to treat severe chronic heart failure	Chronic LV dysfunction LVEF 20–45% 35–75 years old NYHA Class III-IV	Intramyocardial injection of allogeneic iPSC-CMs at time of CABG surgery	20	May 2019	China
NCT 04696328 None-Masking Single-Arm Trial	Cuorips Inc., Osaka University Hospital	Clinical trial of human (allogeneic) iPS cell-derived cardiomyocytes sheet for ischemic cardiomyopathy	Ischemic cardiomyopathy LVEF ≤ 35% ≥20 years old NYHA Class III-IV	Transplantation of human (allogeneic) iPSC-CM Sheet	8	December 2019	Japan
NCT 04396899 None-Masking Single-Arm Trial Phase I/II	Repairon, University Medical Center Goettingen	Safety and efficacy of induced pluripotent stem cell-derived engineered human myocardium as biological ventricular assist tissue in terminal heart failure	HFrEF LVEF < 35% 18–80 years old NYHA Class III-IV	Implantation of Engineered Human Myocardium (EHM) on dysfunctional left or right ventricular myocardium	53	February 2020	Germany
jRCTa 032200189 None-Masking Single-Arm Trial Phase I/II	Heartseed Inc., Keio University School of Medicine	Safety study of induced pluripotent stem cell-derived cardiac spheres transplantation (IPSCS study)	HFrEF LVEF 15–40% 20–75 years old NYHA Class III-IV	Intramyocardial injection of iPSC-CMs spheroids at time of CABG surgery	10	November 2020	Japan

## Data Availability

Not applicable.

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
