# Peer review of "Current Status of Cardiac Regenerative Therapy Using Induced Pluripotent Stem Cells"

_ijms, 2024, doi:10.3390/ijms25115772_

Round 1
Reviewer 1 Report
Comments and Suggestions for Authors
A review paper entitled "Current status of cardiac regenerative therapy using induced pluripotent stem cells" describes the current state of cardiac regeneratice therapy using iPS cells.
Most of the literature used, includes publications from the last 10 years. Although the work is written in an understandable and transparent way, it requires some minor changes.
1. You need to fill in the affiliation next to the names.
2. No superscripts when writing the number of the cells e.g. line 252: 3.3 x107 insted 3.3 x 107. Same thing, lines 314, 315.
3. The abstract must be changed to reflect what the work is about.
4. To make the work more valuable, it is worth adding section 2.1 describing the use of iPS cells in cardiological research at the in vitro and in vivo level.
Author Response
Thank you for your suggestions. We have revised our manuscript as follows.
1 You need to fill in the affiliation next to the names.
We have revised as follows.
Tadahisa Sugiura, MD, PhD1, Dhienda C Shahannaz, MD1, Brandon E Ferrell, MD1
1 Department of Cardiothoracic and Vascular Surgery, Montefiore Medical Center/Albert Einstein College of Medicine, NY, USA
2. No superscripts when writing the number of the cells e.g. line 252: 3.3 x107 insted 3.3 x 107. Same thing, lines 314, 315.
We have revised superscripts.
3. The abstract must be changed to reflect what the work is about.
We have added following text in the abstract.
In this review, we discuss the following topics.
- What is iPSCs
2. Limitations and solutions for translation of iPSC-CMs practically
3. Current therapeutic clinical trials
4. To make the work more valuable, it is worth adding section 2.1 describing the use of iPS cells in cardiological research at the in vitro and in vivo level.
Thank you for your suggestion. We have not added section 2.1, but we have added comments on in vitro and in vivo research and added subsections in Section 2 according to another reviewer's suggestion.
Reviewer 2 Report
Comments and Suggestions for Authors
The review of Sugiura et al., gives a short and concise overview on cardiac regenerative therapy. I think that this topic, as well as this review, are of interest for the field since the first clinical trial results have been published and are discussed here in this review. In general, I think that the authors can implement more original article citations rather than other reviews on that topic. Besides that please see my minor comments below.
Minor comments:
· Double-check the author list to indicate who contributed equally to the manuscript.
· Line 109-110: Please integrate citation for the statement “25-30% binucleated”.
· Line 167: Change “into” to “in”.
· Line 167: Change “&” to “and”.
· Line 173-177: Pleas rephrase this paragraph to make it clearer.
· Erase paragraph “3.5. Solutions for arrhythmogenesis”. Paragraph 3.4. is the same. Adapt the new numbering throughout the manuscript.
· Line 252: Make sure that the numbers are depicted correctly “3.3 × 107 cells”.
· Line 279: Please add what “200μm” means – diameter, radius?
· Line 305: Check spacing “The study demonstrated:1) Safe”.
Comments on the Quality of English LanguageMinor editing necessary.
Author Response
Thank you for your suggestions. We have revised our manuscript as follows.
Double-check the author list to indicate who contributed equally to the manuscript.
We have removed it.
· Line 109-110: Please integrate citation for the statement “25-30% binucleated”.
We have added the references.
· Line 167: Change “into” to “in”.
We have revised it.
· Line 167: Change “&” to “and”.
We have revised it.
· Line 173-177: Pleas rephrase this paragraph to make it clearer.
We have revised these sentences as follows. "iPSCs carrying knockout mutations for two key components (β2 microglobulin and class II MHC class transactivator) of MHC I and II (i.e. HLA I/II knockout iPSCs) were generated using CRISPR/Cas9 gene editing system and differentiated in CMs."
· Erase paragraph “3.5. Solutions for arrhythmogenesis”. Paragraph 3.4. is the same. Adapt the new numbering throughout the manuscript.
We have revised it.
· Line 252: Make sure that the numbers are depicted correctly “3.3 × 107 cells”.
We have revised it.
· Line 279: Please add what “200μm” means – diameter, radius?
We have added "diameter".
· Line 305: Check spacing “The study demonstrated:1) Safe”.
We have revised it.
Reviewer 3 Report
Comments and Suggestions for Authors
The authors present a narrative review on the current status of cardiac regenerative therapy using induced pluripotent stem cells (iPSC). The authors discuss historical context, applications, advantages, challenges for translation to clinical practice, and some ongoing clinical trials. The topic is interesting. Suggestions:
1) In the abstract: please add details about what will be covered in the review article (e.g. may be list out all subheadings from the main text).
2) In line 89 it is mentioned that iPSC have 3 major applications: disease modeling, screening, and regenerative therapies: consider making this a separate section with some specific details for each of the three applications.
3) Lines 102-103 talk about challenges for production into the "public market": please rephrase these lines to clarify that these limitations are for upscaling the production for research; iPSC are not ready for routine clinical/public use.
4) Please mention limitations for translation to clinical practice.
5) Please consider making a table of ongoing clinical trials, with details such as trial registration, trial design (RCT, nonRCT, cross-over, single arm phase II trial etc), country, target population, sample size, intervention and comparator, and the outcomes being measured.
Comments on the Quality of English Languagen/a
Author Response
Thank you for your suggestions. We have revised our manuscript following your suggestions.
1) In the abstract: please add details about what will be covered in the review article (e.g. may be list out all subheadings from the main text).
We have added following text in the abstract.
In this review, we discuss the following topics.
- What is iPSCs
2. Limitations and solutions for translation of iPSC-CMs practically
3.Current therapeutic clinical trials
2) In line 89 it is mentioned that iPSC have 3 major applications: disease modeling, screening, and regenerative therapies: consider making this a separate section with some specific details for each of the three applications.
We have added those separate sections in Section 2.
3) Lines 102-103 talk about challenges for production into the "public market": please rephrase these lines to clarify that these limitations are for upscaling the production for research; iPSC are not ready for routine clinical/public use.
We have revised this sentence as below.
There are still some obstacles in the scalable production of iPSC-CMs.
4) Please mention limitations for translation to clinical practice.
We have already mentioned limitations for translation to clinical practice in the section" 3. Limitations and solutions for translation of iPSC-CMs practically".
5) Please consider making a table of ongoing clinical trials, with details such as trial registration, trial design (RCT, nonRCT, cross-over, single arm phase II trial etc), country, target population, sample size, intervention and comparator, and the outcomes being measured.
We have added the table.